

# Genomic and functional analysis of *Romboutsia ilealis* CRIB[T] reveals adaptation to the small intestine

Jacoline Gerritsen[1,2,*], Bastian Hornung[1,3,*], Bernadette Renckens[4], Sacha A.F.T. van Hijum[4,5], Vitor A.P. Martins dos Santos[3,6], Ger T. Rijkers[7,8], Peter J. Schaap[3], Willem M. de Vos[1,9] and Hauke Smidt[1]

[1] Laboratory of Microbiology, Wageningen University & Research, Wageningen, The Netherlands
[2] Winclove Probiotics, Amsterdam, The Netherlands
[3] Laboratory of Systems and Synthetic Biology, Wageningen University & Research, Wageningen, The Netherlands
[4] Nijmegen Centre for Molecular Life Sciences, CMBI, Radboud UMC, Nijmegen, The Netherlands
[5] NIZO, Ede, The Netherlands
[6] LifeGlimmer GmbH, Berlin, Germany
[7] Laboratory for Medical Microbiology and Immunology, St. Antonius Hospital, Nieuwegein, The Netherlands
[8] Department of Science, University College Roosevelt, Middelburg, The Netherlands
[9] Departments of Microbiology and Immunology and Veterinary Biosciences, University of Helsinki, Helsinki, Finland
[*] These authors contributed equally to this work.

Corresponding author
Bastian Hornung,
bastian.hornung@gmx.de

## ABSTRACT

**Background**. The microbiota in the small intestine relies on their capacity to rapidly import and ferment available carbohydrates to survive in a complex and highly competitive ecosystem. Understanding how these communities function requires elucidating the role of its key players, the interactions among them and with their environment/host.

**Methods**. The genome of the gut bacterium *Romboutsia ilealis* CRIB[T] was sequenced with multiple technologies (Illumina paired-end, mate-pair and PacBio). The transcriptome was sequenced (Illumina HiSeq) after growth on three different carbohydrate sources, and short chain fatty acids were measured via HPLC.

**Results**. We present the complete genome of *Romboutsia ilealis* CRIB[T], a natural inhabitant and key player of the small intestine of rats. *R. ilealis* CRIB[T] possesses a circular chromosome of 2,581,778 bp and a plasmid of 6,145 bp, carrying 2,351 and eight predicted protein coding sequences, respectively. Analysis of the genome revealed limited capacity to synthesize amino acids and vitamins, whereas multiple and partially redundant pathways for the utilization of different relatively simple carbohydrates are present. Transcriptome analysis allowed identification of the key components in the degradation of glucose, L-fucose and fructo-oligosaccharides.

**Discussion**. This revealed that *R. ilealis* CRIB[T] is adapted to a nutrient-rich environment where carbohydrates, amino acids and vitamins are abundantly available.

## INTRODUCTION

Intestinal microbes live in a complex and dynamic ecosystem, and to survive in this highly competitive environment, they have developed close (symbiotic) associations with a diverse array of other intestinal microbes and with their host. This has led to a complex network of host-microbe and microbe-microbe interactions in which the intestinal microbes and the host co-metabolise many substrates (*Backhed et al., 2005*; *Scott et al., 2013*). In addition to competition for readily available carbohydrates in the diet, intestinal microbes are able to extract energy from dietary polysaccharides that are indigestible by the host (*Flint et al., 2012*). Furthermore, intestinal microbes can utilize host-derived secretions (e.g., mucus) as substrates for metabolic processes (*Ouwerkerk, De Vos & Belzer, 2013*). In turn, the metabolic activities of the intestinal microbes result in the production of a wide array of compounds, of which some are important nutrients for the host. For example, short chain fatty acids (SCFA), the main end-products of bacterial fermentation in the gut, can be readily absorbed by the host and further metabolized as energy sources (*Elia & Cummings, 2007*; *Lange et al., 2015*). All together, the metabolic activity of the intestinal microbiota has a major impact on the health of the host, and recent studies have indicated an important role for microbial activity in diseases such as inflammatory bowel disease, irritable bowel syndrome and obesity (*Gerritsen et al., 2011a*; *Quigley, 2013*).

We only have a limited understanding of the heterogeneity in microbial community composition and activity in different niches along the length of the intestinal tract. To unravel the functional contribution of specific intestinal microbes to host physiology and pathology, we have to understand their metabolic capabilities at a higher resolution. It is still difficult, however, to associate a functionality in this ecosystem to specific sets of genes and in turn to individual microbial species, and vice versa. To this end, the combination of genome mining and functional analyses with single microbes or with simple and defined communities can provide an overall insight in the genetic and functional potential of specific members of the intestinal microbial community (*Heinken et al., 2013*; *Li et al., 2008*; *Xu et al., 2003*).

As mentioned above, intestinal microbes have adapted or even specialized in foraging certain niche-specific substrates. However, little is known about the adaption of intestinal microbes to the conditions in the small intestine (*Booijink et al., 2007*; *Van den Bogert et al., 2013b*; *Zhang et al., 2014*). Community composition and activity in the small intestine is largely determined by the host digestive fluids such as gastric acid, bile and pancreatic secretions. The small intestine is a nutrient-rich environment, and previous studies have shown that the microbial communities in the (human) small intestine are driven by the rapid uptake and conversion of simple carbohydrates (*Zoetendal et al., 2012*; *Leimena et al., 2013*). Genomic studies of small intestinal isolates have indicated environment-specific adaptations to the small intestine with respect to their carbohydrate utilization capacities, which was evidenced by the presence of a wide array of genes involved in nutrient transport and metabolism of, mainly simple, carbohydrates (*Van den Bogert et al., 2013a*).

Here we describe a model driven genomic analysis of the small intestinal inhabitant *Romboutsia ilealis* CRIB[T] (*Gerritsen et al., 2014*). *R. ilealis* CRIB[T] is currently still the

only isolate of the recently descibed species *R. ilealis*, a species that belongs to the family *Peptostreptococcaceae,* of which many members are common intestinal microbes including the well-known species *Clostridioides difficile* (previously known as *Clostridium difficile*) and *Intestinibacter bartlettii* (previously known as *Clostridum bartlettii*) (*Galperin et al., 2016*). An overview of the metabolic capabilities and nutritional potential of the type strain of *R. ilealis* CRIB^T is provided here to identify potential mechanisms that enable this organism to survive in the competitive small intestinal environment.

## MATERIALS AND METHODS

### Genome sequencing, assembly and annotation

*R. ilealis* CRIB^T (DSM 25109) was routinely cultured in CRIB medium at 37 °C as previously described (*Gerritsen et al., 2014*). Genomic DNA extraction was performed as previously described (*Van den Bogert et al., 2013a*). Genome sequencing was done using 454 Titanium pyrosequencing technology (Roche 454 GS FLX), as well as Illumina (Genome Analyzer II and HiSeq2000) and PacBio sequencing (PacBio RS). Mate-pair data was generated by BaseClear (Leiden, the Netherlands). All other data was generated by GATC Biotech (Konstanz, Germany). The genome was assembled in a hybrid approach with multiple assemblers. In short, after estimation of the genome size, assembly of the genome was performed with two different assemblers in parallel using the different sequence datasets. After merging the two assemblies three rounds of scaffolding were performed, once with paired-end data and twice with mate-pair data. Gap-filling was performed after each scaffolding step.

Genome annotation was carried out with an in-house pipeline. Prodigal v2.5 was used for prediction of protein coding DNA sequences (CDS) (*Hyatt et al., 2010*), InterProScan 5RC7 for protein annotation (*Hunter et al., 2012*), tRNAscan-SE v1.3.1 for prediction of tRNAs (*Lowe & Eddy, 1997*) and RNAmmer v1.2 for the prediction of rRNAs (*Lagesen et al., 2007*). Additional protein function predictions were derived via BLAST identifications against the UniRef50 (*Suzek et al., 2007*) and Swissprot (*UniProt-Consortium, 2014*) databases (download August 2013). Afterwards the annotation was further enhanced by adding EC numbers via PRIAM version March 06, 2013 (*Claudel-Renard et al., 2003*). Non-coding RNAs were identified using rfam_scan.pl v1.04, on release 11.0 of the RFAM database (*Burge et al., 2013*). CRISPRs were annotated using CRISPR Recognition Tool v1.1 (*Bland et al., 2007*).

Qualitative metabolic modelling has been performed with Pathway tools v18.0 (*Latendresse et al., 2012*). A generic default medium consisting out of ammonia/urea, sulfite, hydrogen sulfide and phosphate was assumed, and the qualitative possibility to produce all necessary biomass metabolites was tested with the supply of different carbohydrates, which had been tested before *in vitro*.

See the Supplemental Methods in Text S1 for details on the genomic DNA extraction, genome sequencing, assembly, annotation, and metabolic modelling.

## Whole-genome transcriptome analysis

*R. ilealis* CRIB$^T$ was grown in a basal bicarbonate-buffered medium (*Stams et al., 1993*) supplemented with 16 g/L yeast extract (BD, Breda, The Netherlands) and an amino acids solution as used for the growth of *C. difficile* (*Karasawa et al., 1995*). In addition, the medium was supplemented with either 0.5% (w/v) D-glucose (Fisher Scientific Inc., Waltham, MA USA), L-fucose (Sigma-Aldrich, St. Louis, MO, USA) or fructo-oligosaccharide (FOS) P06 (DP 2-4; Winclove Probiotics, Amsterdam, The Netherlands). The final pH of the medium was adjusted to 7.0. For each condition, triplicate cultures were set up. For RNA-seq analysis, the cells were harvested in mid-exponential phase ($OD_{600 nm} = 0.25$–0.55, ~8–10 h incubation) (Table S1).

Total RNA was purified using the RNeasy Mini Kit (QIAGEN GmbH, Hilden, Germany). Depletion of rRNA was performed using the Rib-Zero$^{TM}$ Kit for bacteria (Epicentre Biotechnologies, Madison, WI, USA). The ScriptSeq$^{TM}$ v2 RNA-seq Library Preparation Kit in combination with ScriptSeq$^{TM}$ Index PCR primers (Epicentre Biotechnologies) was used for library construction for whole-transcriptome sequencing (RNA-seq). The barcoded cDNA libraries were pooled and sent to GATC Biotech (Konstanz, Germany) where 150 bp sequencing was performed on one single lane using the Illumina HiSeq2500 platform in combination with the TruSeq Rapid SBS (200 cycles) and TruSeq Rapid SR Cluster Kits (Illumina Inc., San Diego, CA, USA). Reads were mapped to the genome with Bowtie2 v2.0.6 (*Langmead & Salzberg, 2012*) using default settings, after quality control (rRNA removal, adapter trimming, and quality trimming) had been performed. Details on the RNA-seq raw data analysis can be found in Table S2 and Supplemental Methods in Text S1.

Gene expression abundance estimates and differential expression analysis was performed using Cuffdiff v2.1.1 (*Trapnell et al., 2013*) with default settings. Differentially expressed genes were determined by pairwise comparison of a given condition to the other three conditions for a total of six pairwise comparisons. Genes were considered significantly differentially expressed when they showed a $\geq 1.5 \log2$ (fold change) in any of the conditions with a false discovery rate (FDR)-corrected $P$ value ($q$ value) $\leq 0.05$ (Tables S3–S6). Principal component analysis was performed with Canoco 5.0 (*Ter Braak & Smilauer, 2012*) on log-transformed gene transcript abundances using Hellinger standardization. Gene expression heatmaps were generated based on gene transcript abundances using R v3.1.0 and R-packages svDialogs and gplots.

See the Supplemental Methods in Text S1 for details on growth on different carbohydrate media and whole genome transcriptome analysis.

## Metagenomic investigations

The datasets PRJNA237362 (*Gevers et al., 2014*) and PRJNA298762 (*Alipour et al., 2016*) were analysed as relevant representative publicly available 16s rRNA gene amplicon datasets for the presence of 16S rRNA gene sequences closely related to that of *R. ilealis* with NG-Tax version 0.3 (*Ramiro-Garcia et al., 2016*) with the–classifyRatio argument set to 0.9.

**Table 1  General features of the *R. ilealis* CRIB^T genome.**

|  | Chromosome | Plasmid |
|---|---|---|
| Size (bp) | 2,581,778 | 6,145 |
| G + C content (%) | 27.9 | 29.3 |
| Protein CDS | 2,351 | 8 |
|    Pseudogenes | 12 | 0 |
| Coding density | 1.10 | 1.02 |
| Average gene size (bp) | 899 | 531 |
| rRNA genes |  |  |
|    16S rRNA genes | 14 | 0 |
|    23S rRNA genes | 14[a] | 0 |
|    5S rRNA genes | 14 | 0 |
| tRNAs | 109 | 0 |
| ncRNAs | 28 | 0 |
| CRISPR repeats | 1[a]71 | 0 |

**Notes.**

[a] An additional 23S rRNA gene is expected in one of the gaps.

## Nucleotide sequence accession number

All related data have been deposited in the European Nucleotide Archive. The raw reads for the genome of *R. ilealis* CRIB^T can be accessed via the accession numbers ERR366773, ERX397233, ERX397242 and ERX339449. The assembly can be accessed under LN555523–LN555524. The RNAseq data have been deposited under the numbers ERS533849–ERS533860.

# RESULTS

## Genome analysis

### Global genome features

*R. ilealis* CRIB^T contains a single, circular chromosome of 2,581,778 bp and a plasmid of 6,145 bp (Table 1 and Fig. 1). The chromosome contains 2,351 predicted protein CDS, of which 321 were annotated as hypothetical and for 91, only a domain of unknown function could be assigned. The plasmid carries eight predicted protein CDS, of which none was recognized for having a metabolic or replicative function. Furthermore, it appears to be a non-mobilizable plasmid, given that it lacks any known mobilization-associated genes. The overall G + C content of the genome is 27.9%, which is in good agreement with a G + C content of 28.1 mol% previously determined for *R. ilealis* CRIB^T by HPLC methods (*Gerritsen et al., 2014*).

With a total of 14 copies of the 16S ribosomal RNA (rRNA) gene, *R. ilealis* CRIB^T is among the species with the highest number of 16S rRNA gene copies reported up to this date (*Lee, Bussema & Schmidt, 2009*). High numbers of rRNA operons have been proposed to be indicative for fast growth and to allow microbes to respond quickly to changes in available resources (*Klappenbach, Dunbar & Schmidt, 2000*). In addition, a high copy number of the rRNA operon has been suggested to be essential for successful sporulation and germination (*Yano et al., 2013*). This is also reflected in the observation that in general

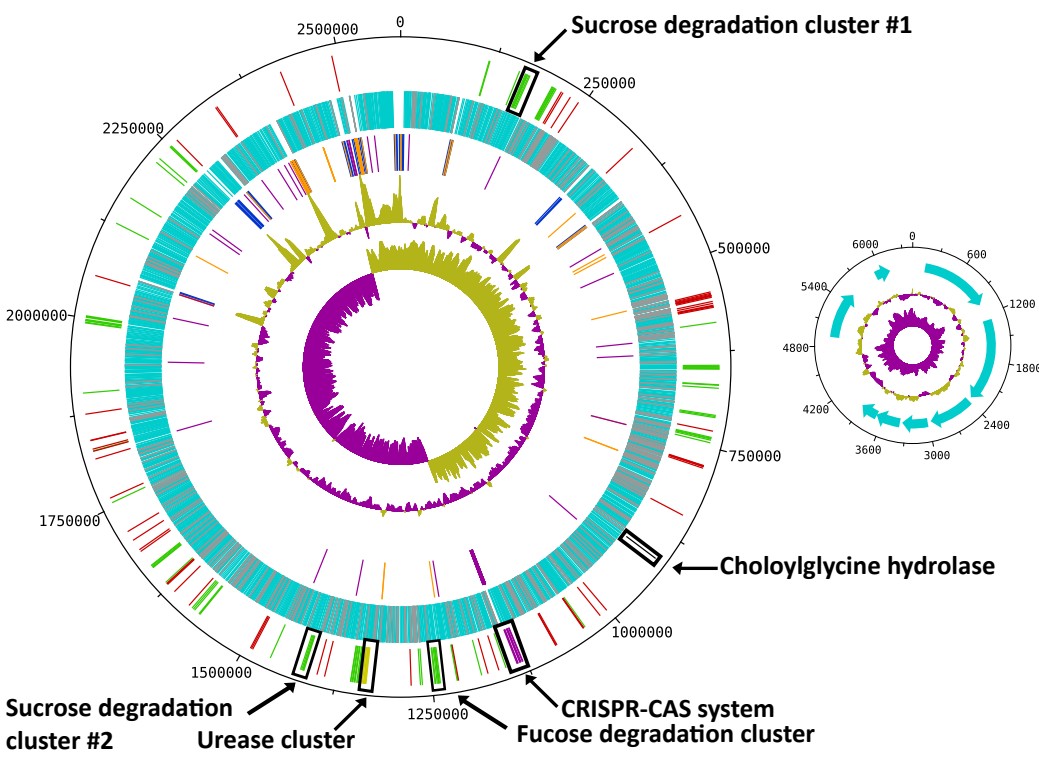

**Figure 1  Circular map of the *R. ilealis* CRIB[T] genome.** Both chromosome and non-mobilizable plasmid are shown. For the chromosome tracks from inside to outside are as follows: 1, GC skew; 2, G + C content; 3, RNAs [rRNAs (blue), tRNAs (orange) and ncRNAs (purple)]; 4, all predicted protein CDS [with predicted function (light-blue), hypothetical proteins and proteins to which only a domain of unknown function could be assigned (grey)]; 5, genes or gene clusters of interest [(mobile genetic elements (red), Cas proteins (pink), urease gene cluster (yellow), choloylglycine hydrolase (black), gene clusters involved in carbohydrate utilization (green)]. For the plasmid tracks from inside to outside are as follows: 1, GC skew; 2, G + C content; 3, all predicted CDS.

the species that contain the highest number of reported rRNA operons, including *R. ilealis* CRIB[T], belong to the spore-forming bacterial orders *Bacillales* and *Clostridiales*. Not all of the 16S rRNA gene copies in *R. ilealis* CRIB[T] are embedded in the conserved 16S-23S-5S rRNA operon structure. Of the fifteen locations containing rRNA genes, ten are in the classical order 16S-23S-5S. The other five operons are characterized by duplicated or missing rRNA genes, or a different order of the genes. It should be noted that the current assembly contains three gaps, all of which are located within rRNA operons. Diverging rRNA operon structures have been reported for other genomes containing multiple rRNA operons, as a result of duplications (*Bensaadi-Merchermek et al., 1995*; *Schwartz, Gazumyan & Schwartz, 1992*).

A cluster of orthologous genes (COG) category (*Tatusov, Koonin & Lipman, 1997*) could be assigned to 1,647 of the predicted proteins (70%) including 372 proteins (16%) assigned to the categories R (general function prediction only) and S (function unknown) (Fig. S1). With InterProScan a predicted function could be assigned to 82% of the predicted

proteins. Based on the InterPro and PRIAM classifications (*Claudel-Renard et al., 2003*), an enzymatic function could be predicted for more than 500 proteins.

### General metabolic pathways

Analysis of the CDS predicted from the *R. ilealis* CRIB$^T$ genome revealed the presence of a complete set of enzymes for the glycolytic pathway. In line with the anaerobic lifestyle of the organism, enzymes for the oxidative phase of the pentose phosphate pathway could not be detected. Additionally, the genes that encode enzymes involved in the tricarboxylic acid cycle were lacking. Subsequently a metabolic model was constructed with Pathway tools v18.0. A flux balance analysis with the model was performed, suggesting that *R. ilealis* CRIB$^T$ is a mixed acid fermenter as previously reported (*Gerritsen et al., 2014*). Predicted end products of fermentation are a mixture of acetate, formate, lactate and ethanol, with the possibility of gas formation ($CO_2$ and $H_2$). In addition to ethanol, which can be produced during mixed acid fermentation, 1,2-propanediol was predicted to be formed via the L-fucose degradation pathway. The fermentation end products formate, acetate and lactate are predicted to be produced from pyruvate. No other solvents were predicted to be produced by the metabolic model. The only metabolite produced by *R. ilealis* CRIB$^T$ that was not accounted for by the metabolic model, was propionate. None of the three established pathways for propionate production in the intestinal tract, i.e., the succinate, acrylate or the propanediol pathway (*Reichardt et al., 2014*), could be identified at the genetic level in the genome of *R. ilealis* CRIB$^T$. Although propionate is only produced in low amounts (max. 3 mM in 24 h) it is noteworthy because propionate production was observed repeatedly during *in vitro* growth in this study (see Table 2) and as previously reported (*Gerritsen et al., 2014*).

The analysis of the genome and the prediction by the model indicated that fermentation is probably the main process for energy conservation in *R. ilealis*. However, the presence of a sulfite reductase gene cluster (CRIB_1284-CRIB_1286) of the dissimilatory *asrC*-type (*Dhillon et al., 2005*) points at possible anaerobic respiration. Similar siroheme-dependent sulfite reductases are found in many close-relatives of *R. ilealis* such as *I. bartlettii*, *Clostridium sordellii* and *C. difficile* (*Czyzewski & Wang, 2012*). Sulfite reduction by *R. ilealis* CRIB$^T$, and close relatives, has been previously demonstrated *in vitro* (*Gerritsen et al., 2014*), and increased growth yield and metabolite production was observed in the presence of sulfite for *R. ilealis* CRIB$^T$ (Table S7). In the intestinal tract, sulfite is derived from food sources that contain sulfite as a preservative, and it has been shown that neutrophils release sulfite as a part of the host defence against microbes (*Mitsuhashi et al., 1998*).

### Metabolism of growth factors and cofactors

Complete pathways are present for the biosynthesis of the amino acids aspartate, asparagine, glutamate, glutamine and cysteine, using carbon skeletons available from central metabolites or via conversion of other amino acids. However, many genes encoding enzymes required for biosynthesis of other amino acids appeared to be absent in *R. ilealis* CRIB$^T$. As most missing genes are part of well-studied pathways, it is unlikely these functionalities are encoded by unknown genes and likely represent true auxotrophies. The absence of genes to produce branched-chain amino acids (leucine, isoleucine and valine)

Gerritsen et al. (2017), *PeerJ*, DOI 10.7717/peerj.3698

**Table 2  Fermentation end products of *R. ilealis* CRIB[T] produced during growth on different carbohydrates (glucose, FOS or L-fucose) or in basal medium in the absence of a carbon source (control condition).** Samples were obtained during mid-exponential phase (∼8–10 h incubation; used for transcriptome analysis) and in stationary phase (24 h incubation). For the control cultures, fermentation products are shown for the individual cultures separating the carbohydrates used for preconditioning of the inoculum. For the three other conditions, values represent means of triplicate cultures with standard deviations.

| | Formate (mM) | | Acetate (mM) | | Propionate (mM) | | Lactate (mM) | | 1,2-propane-diol (mM) | |
|---|---|---|---|---|---|---|---|---|---|---|
| | 8–10 h | 24 h | 8–10 h | 24 h | 8–10 h | 24 h | 8–10 h | 24 h | 8–10 h | 24 h |
| Control: basal medium | | | | | | | | | | |
| (glucose inoc.) | 3.2 | 7.7 | 2.0 | 6.2 | 2.0 | 2.2 | N.D. | N.D. | N.D. | N.D. |
| (FOS inoc.) | 4.5 | 9.2 | 2.4 | 7.4 | 2.4 | 2.9 | N.D. | N.D. | N.D. | N.D. |
| (L-fucose inoc) | 4.8 | 10.8 | 2.3 | 9.8 | 2.3 | 3.0 | N.D. | N.D. | 1.0 | 1.0 |
| Basal medium + glucose (5% w/v) | 4.4 ± 1.2 | 28.2 ± 4.3 | 1.0 ± 0.9 | 16.3 ± 2.2 | 1.0 ± 0.9 | 1.3 ± 0.1 | N.D. | 3.0 ± 0.7 | N.D. | N.D. |
| Basal medium + FOS (5% w/v) | 4.7 ± 0.6 | 27.3 ± 2.5 | 1.4 ± 0.0 | 17.7 ± 1.4 | 1.4 ± 0.0 | 1.6 ± 0.1 | N.D | 2.5 ± 0.3 | N.D. | N.D. |
| Basal medium + L-fucose (5% w/v) | 6.7 ± 0.1 | 19.5 ± 3.6 | 2.8 ± 0.1 | 16.3 ± 2.9 | 2.8 ± 0.1 | 2.8 ± 0.4 | N.D. | N.D. | 1.3 ± 0.1 | 7.7 ± 1.4 |

**Notes.**
N.D., not detected.

was also reflected in the absence of branched chain fatty acids in the cell membrane of *R. ilealis*, which is characteristic for the genus *Romboutsia* (*Gerritsen et al., 2014*). From these observations it can be concluded that *R. ilealis* depends on a number of exogenous amino acids, peptides and/or proteins to fuel protein synthesis. The dependency on an exogenous source of amino acids is reflected by the identification of multiple amino acid transporters, including an arginine/ornithine antiporter, multiple serine/threonine exchangers, a transporter for branched amino acids, and several amino acid symporters and permeases without a predicted specificity. Furthermore, numerous genes were annotated as protease or peptidase, including several with a signal peptide.

*R. ilealis* CRIB[T] appears to contain all genes for *de novo* purine and pyrimidine synthesis, as well as for the production of the coenzymes NAD and FAD via salvage pathways from niacin and riboflavin, respectively. While some organic cofactors can be produced by *R. ilealis* CRIB[T], it mainly relies on salvage pathways (e.g., for lipoic acid) or exogenous sources for the supply of precursors, mainly in the form of vitamins (e.g., thiamin, riboflavin, niacin, pantothenate, pyridoxine, biotin, vitamin B12).

### Carbohydrate transport and metabolism

As previously reported, *R. ilealis* CRIB[T] is able to utilize a wide variety of carbohydrates (*Gerritsen et al., 2014*). Previously, good growth of *R. ilealis* on L-fucose, glucose, raffinose and sucrose was described, in addition to moderate growth on D-arabinose and D-galactose and weak growth on D-fructose, inulin, lactose, maltose and melibiose. Growth on L-fucose, fructose, galactose, glucose, lactose, maltose, melibiose, raffinose and sucrose was predicted from the genome-scale metabolic model as well. For these different carbohydrates, the genes encoding the specific carbohydrate degradation enzymes were found distributed throughout the genome in gene clusters together with their respective transporters and transcriptional regulator. The only carbohydrate utilized by *R. ilealis* CRIB[T] that was not predicted based on the metabolic model, was D-arabinose. Although a separate arabinose transporter, similar to the maltose and sucrose transporters, could be identified in the genome *R. ilealis* CRIB[T], no separate pathway for the use of D-arabinose could be predicted, However, it is likely that the L-fucose degradation pathway (encoded by genes CRIB_1294-CRIB_1298) is also used for D-arabinose utilization as is also observed in other intestinal species (*LeBlanc & Mortlock, 1971*). In addition to the carbohydrates for which growth was studied, a gene cluster involved in the degradation of the host-derived carbohydrate sialic acid could be predicted (CRIB_613-CRIB_619) (*Almagro-Moreno & Boyd, 2009*). The structure of this gene cluster is similar to the one identified in *C. difficile* (*Ng et al., 2013*). The ability to degrade the predominantly host-derived carbohydrates, L-fucose and sialic acid, suggest a role in the utilization of mucin, an abundant host-derived glycoprotein in the intestinal tract (*Derrien et al., 2010*; *Ouwerkerk, De Vos & Belzer, 2013*). However, no growth on mucin was observed (Table S7), which is in line with the lack of a predicted extracellular fucosidase and/or sialidase.

### Other genes encoding niche-specific functionalities

A gene cluster encoding a urease, consisting of three subunits (*ureABC*), and a number of urease accessory genes was identified (CRIB_1381-CRIB_1388). The gene cluster identified

in *R. ilealis* CRIB[T] is very similar to the urease gene cluster in the genome of *C. sordellii* (Fig. S3), a species in which the urease activity is used to phenotypically distinguish *C. sordellii* strains from *C. bifermentans* strains (*Roggentin et al., 1985*). Furthermore, a possible ammonium transporter (CRIB_1389) was identified in the genome of *R. ilealis* CRIB[T] next to the urease gene cluster. Ureases are nickel-containing metalloenzymes that catalyse the hydrolysis of urea to ammonia and carbon dioxide, and thereby these enzymes allow microbes to use urea as nitrogen source by assimilation via glutamate. They are ubiquitous proteins occurring in diverse organisms (*Mobley, Island & Hausinger, 1995*). In the intestinal environment, where urea is abundantly present (*Fuller & Reeds, 1998*), some bacteria use ureases to survive the acidic conditions in the upper part of the intestinal tract as urea hydrolysis leads to a local increase in pH (*Rutherford, 2014*).

Another gene encoding a niche-specific functionality is the predicted choloylglycine hydrolase. Proteins within the choloylglycine hydrolase family are bile salt hydrolases (BSHs), also known as conjugated bile acid hydrolases (CBAHs), that are widespread among intestinal microbes (*Ridlon, Kang & Hylemon, 2006*). They are involved in the hydrolysis of the amide linkage in conjugated bile salts, releasing primary bile acids. There is a large heterogeneity among BSHs, for example with respect to their substrate specificity. The BSH of *R. ilealis* CRIB[T] was found to be the most similar to the one found in *Clostridium butyricum*. Although the physiological advantages of BSHs for the microbes are not completely understood, it has been hypothesized that they constitute a mechanism to detoxify bile salts and thereby enhance bacterial colonization (*Czyzewski & Wang, 2012*).

## Metabolite and transcriptome analysis
### Metabolite and transcriptome analysis of R. ilealis CRIB[T] during growth on different carbohydrates

To study key pathways predicted to be involved in carbohydrate utilization and their regulation in more detail, a genome-wide transcriptome analysis was performed, focussing on four experimental conditions. Firstly, growth on glucose, a preferred substrate for many microbes present in the intestinal tract, was studied. Secondly the growth on fructans, oligo- and polysaccharides present in many food items was examined. Previously weak growth on inulin, a polysaccharide consisting of long chains of $\beta1 \rightarrow 2$ linked fructose units, was observed (*Gerritsen et al., 2014*). For this study a shorter fructan (FOS P06, DP2-4) was chosen, because growth on shorter fructans is likely more relevant for microbes living in the small intestine (*Zoetendal et al., 2012*). Thirdly, growth on L-fucose was examined, as growth on this substrate was found to be unique for *R. ilealis* CRIB[T] compared to other related microbes. Finally, *R. ilealis* CRIB[T] was also grown in the basal medium in the absence of an additional carbon source for comparison (control condition).

Based on measurements of optical density and pH during growth (growth characteristics of individual cultures can be found in Table S1), samples were drawn in the mid-exponential phase ($\sim$8–10 h incubation; used for transcriptome analysis) and in stationary phase (24 h incubation), and sugar utilization and fermentation products were measured with HPLC (Table 2). In neither of the experimental conditions the supplied carbohydrates were depleted, and metabolites were still produced at the time of sampling at $\sim$8–10 h and 24 h,

which further confirmed that samples obtained for transcriptome analysis at ∼8–10 h were taken during exponential growth. In the FOS cultures, an accumulation of extra-cellular fructose was observed. As predicted from the metabolic model, growth on glucose resulted in the production of formate, acetate and lactate (Table 2).

Growth on FOS was marginally lower than that on glucose, however, after 24 h of growth, the same fermentation products were observed in similar amounts (Table 2). Growth on L-fucose showed production of 1,2-propanediol instead of lactate. The fact that 1,2-propanediol was observed in one of the control cultures could be explained by the fact that an L-fucose grown culture was used as inoculum for this culture, leading to carry-over of minor amounts of metabolites.

For the genome-wide transcriptome analysis of triplicate cultures grown in the four different conditions (i.e., a total of 12 cultures), a total of 159,250,634 150 bp-reads were generated by RNA-seq (overview in Table S2). Principal component analysis of the transcriptomes of the individual cultures showed that the cultures clustered by condition (Fig. 2).

### Differential expression of genes involved in carbohydrate degradation and fermentation in R. ilealis CRIB[T]

To identify differentially regulated genes, pairwise comparisons were done with cuffdiff (*Trapnell et al., 2013*) using a cut off of $\geq 1.5$ log2 (fold-change) and $q$-value $\leq 0.05$. Figure 3 shows a heat map of all differentially regulated genes, and exact numbers can be found in Tables S3–S6.

The gene cluster involved in glycolysis (CRIB_186-CRIB_191) was most abundantly expressed in the conditions that support the highest growth rates determined by the highest cell density reached in the time period that was measured (glucose, followed by FOS; Fig. 3). This was also reflected in the fact that expression of genes encoding proteins involved in replication such as ribosomal proteins, proteins involved in cell wall biosynthesis and general cell division processes were most strongly expressed during growth in the presence of glucose and to a lesser extent FOS. Other genes involved in the central sugar metabolic pathways (e.g., CRIB_1849, CRIB_140, CRIB_2223, and CRIB_105) were upregulated in these conditions, albeit not significantly differentially regulated. This suggests that these genes are less tightly regulated at the transcriptional level, probably because they are also involved in other processes than sugar degradation (*Commichau et al., 2009*). The metabolic model suggests that this is indeed the case, as some of the enzymes produce intermediates which can be consumed by fatty acid biosynthesis and amino acid biosynthesis processes.

Altogether, the transcriptome of *R. ilealis* CRIB[T] grown on FOS was very similar to its transcriptome when grown on glucose (Fig. 2), with only 18 genes significantly upregulated during growth in the presence of FOS compared to glucose (Table S4). Apparent was the upregulation of the gene clusters that code for proteins involved in the transport and degradation of the respective sugars or their derivatives (Fig. 3). In the presence of glucose, the glucose-specific PTS system (CIRB_2017-CRIB_2018) was significantly upregulated, together with its associated transcriptional regulator (CRIB_2019). In turn, in the presence of FOS, two clusters predicted to be involved in sucrose degradation (CRIB_148-CRIB_152 and CRIB_1458-1461) were significantly upregulated. The third gene cluster predicted to

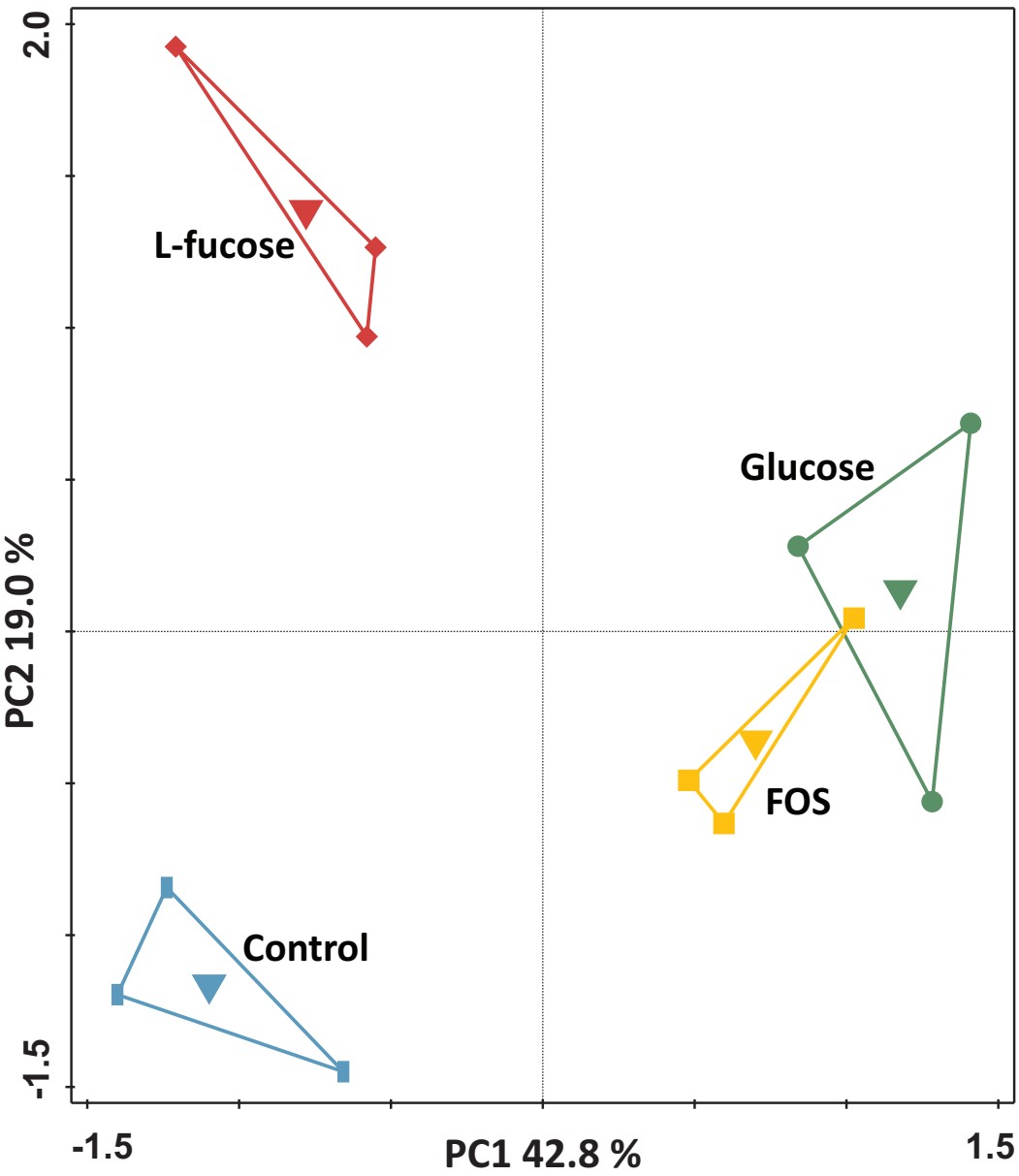

**Figure 2** **Principal component analysis of the transcriptomes of *R. ilealis* CRIB^T grown on different carbohydrates (glucose, FOS and L-fucose) or in the absence of an additional carbon source (control).** First and second ordination axes are plotted, explaining 42.8% and 19.0% of the variability in the data set, respectively. Individual transcriptomes are symbol-coded by experimental condition: glucose (circles), FOS (squares), L-fucose (diamonds) and control (rectangles). The experimental conditions were used as supplementary variables as well and could explain 62.9% of the variation.

be involved in sucrose degradation (CRIB_1399-1400) was not significantly regulated during growth on FOS. However, it should be noted that these genes are located in a cluster functionally annotated to melibiose metabolism and are most likely regulated by the transcriptional regulator in this cluster. In addition to the two sucrose degradation clusters, a transport cluster of unknown function (CRIB_1506-CRIB_1509) was upregulated during

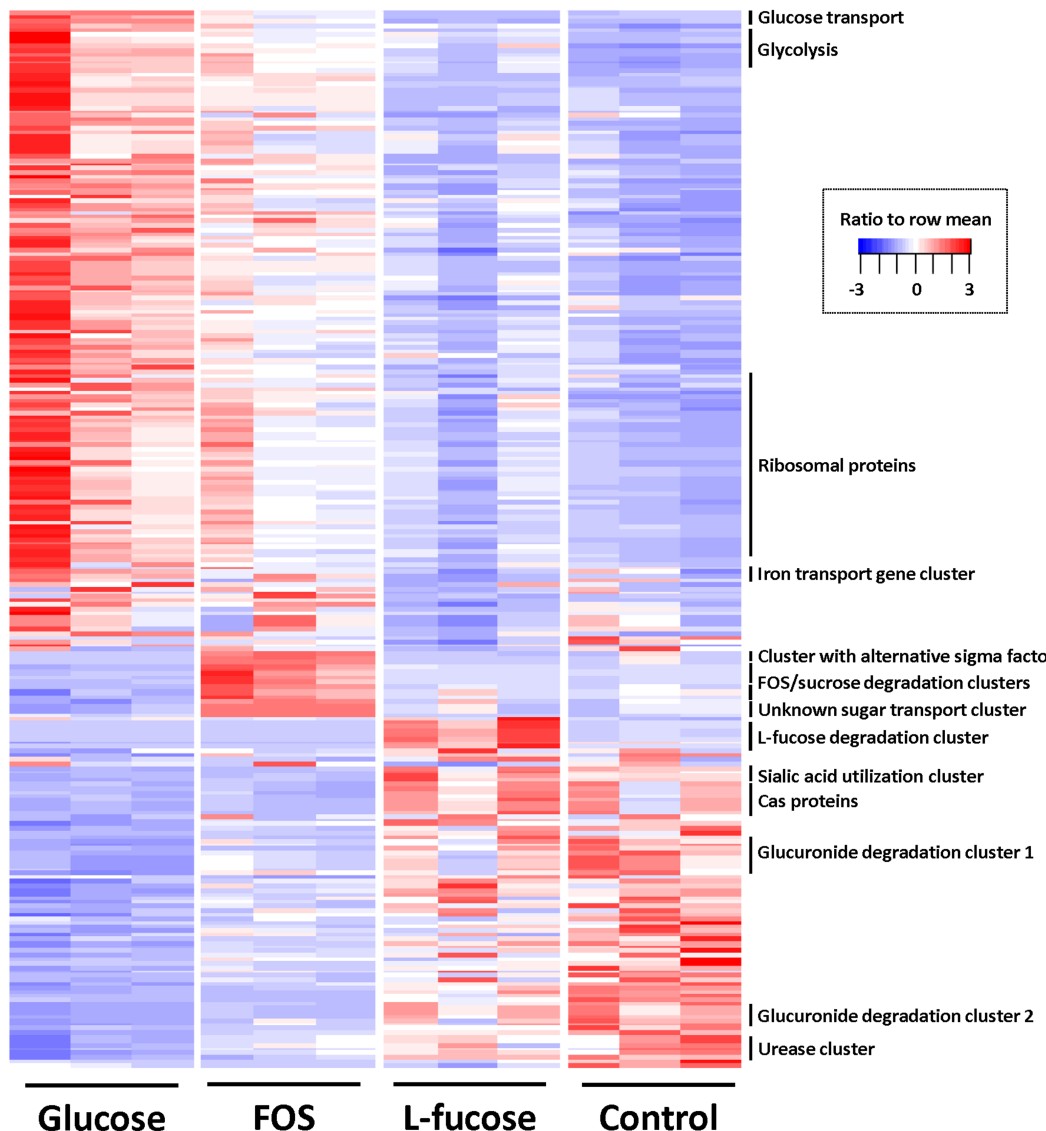

**Figure 3** **Heatmap of genes differentially expressed in at least one of the four conditions ($\geq$ 1.5 log2 (fold change) and $q$ value $\leq$ 0.05).** Colour coding by ratio to row mean. Key gene clusters are indicated.

growth on FOS, albeit only significantly when compared to growth on glucose. During growth in the presence of L-fucose, the gene cluster predicted to be involved in L-fucose degradation (CRIB_1294-CRIB_1298) was significantly upregulated, including the gene encoding the corresponding transcriptional regulator (CRIB_1299). An overview of the main carbohydrate degradation pathways regulated in the different conditions is given in Fig. 4.

During growth on glucose, L-lactate dehydrogenase (CRIB_684) was significantly upregulated, albeit not significantly compared to growth on FOS. This enzyme catalyses the reduction of pyruvate resulting in the production of L-lactate and the reoxidation of the NADH formed during glycolysis. Only at the time point of 24 h, lactate was observed

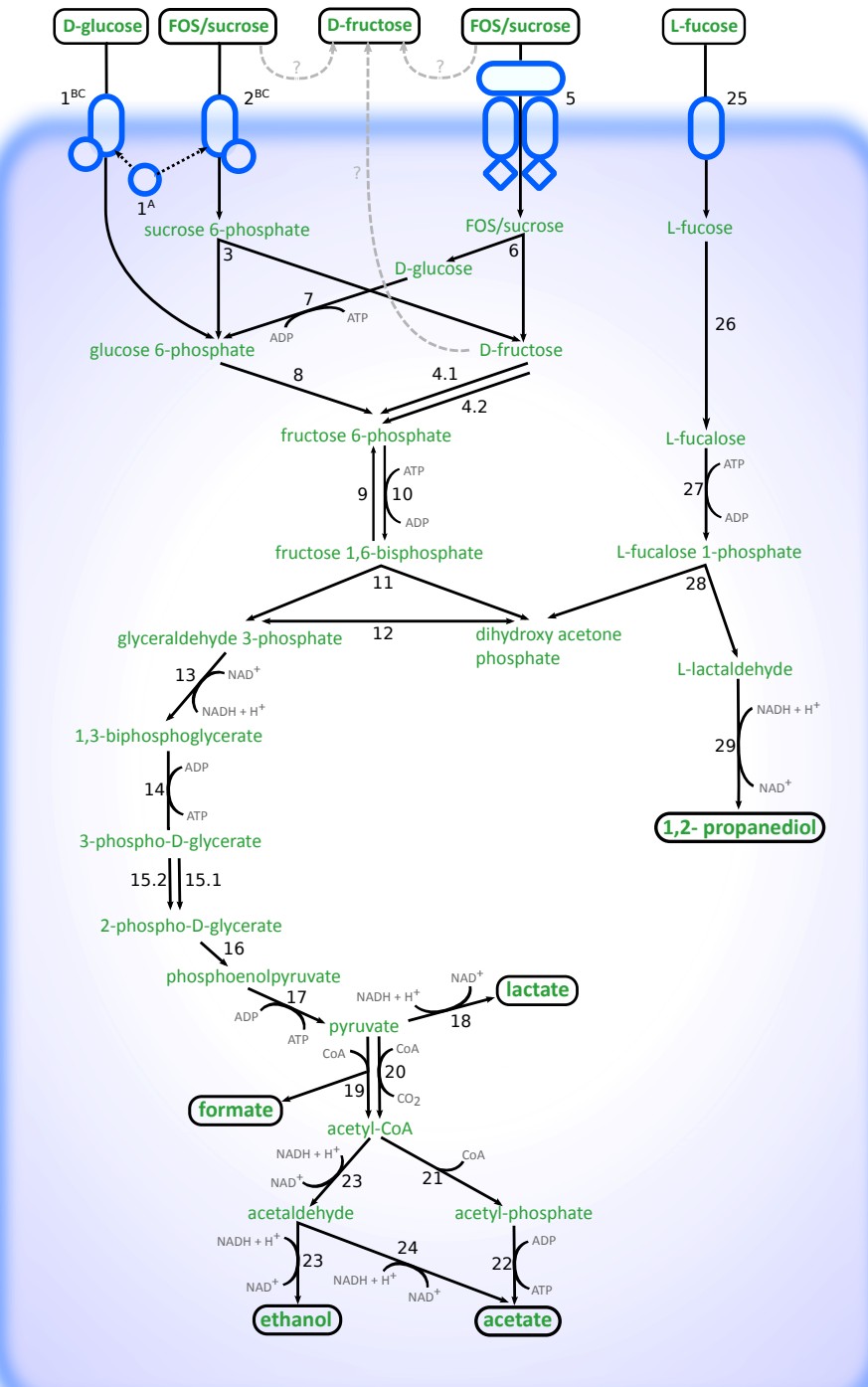

**Figure 4  Schematic overview of the pathways involved in degradation of glucose, FOS and L-fucose in *R. ilealis* CRIB<sup>T</sup>.** 1<sup>A</sup>; PTS system glucose-specific EIIA component (CRIB_2018); 1<sup>BC</sup>, PTS system glucose-specific EIIBC component (CRIB_2017); 2<sup>BC</sup>, PTS system sucrose-specific EIIBC component (CRIB_1461); 3, ß-fructofuranosidase with RDD family protein (CRIB_1459 and CRIB_1460); (continued on next page…)

**Figure 4 (…continued)**
4, fructokinase (CRIB_152 and CRIB_1458); 5; ABC-type transporter (CRIB_148-CRIB_150); 6, ß-fructofuranosidase (CRIB_151); 7, glucokinase (CRIB_1849); 8, glucose 6-phosphate isomerase (CRIB_140); 9, fructose 1,6-bisphosphatase (CRIB_45 and CRIB_2020); 10, 6-phosphofructokinase ; (CRIB_104); 11, fructose-bisphosphate aldolase (CRIB_2223); 12, triosephosphate isomerase (CRIB_189); 13, glyceraldehyde-3-phosphate dehydrogenase (CRIB_187); 14, phosphoglycerate kinase; 15, phosphoglycerate mutase (CRIB_1223) and 2,3-bisphosphoglycerate-independent phosphoglycerate mutase (CRIB_190); 16, enolase (CRIB_191); 17, pyruvate kinase (CRIB_105); 18, L-lactate dehydrogenase (CRIB_684); 19, formate acetyltransferase (CRIB_2141); 20, pyruvate-flavodoxin oxidoreductase (CRIB_2021); 21, phosphate acetyltransferase (CRIB_2171); 22, acetate kinase (CRIB_1927); 23, bifunctional aldehyde-alcohol dehydrogenase (CRIB_2231); 24, fatty aldehyde dehydrogenase (CRIB_2231); 25, L-fucose permease (CRIB_1294); 26, L-fucose isomerase (CRIB_1298); 27, L-fuculokinase (CRIB_1297); 28, L-fuculose phosphate aldolase (CRIB_1297); 29, lactaldehyde reductase (CRIB_1300); ?, possible mechanisms of external fructose accumulation (external degradation, or export).

(Table 2). This suggests that at time point ∼8–10 h the cells were starting to regenerate NAD by upregulating this gene. In the presence of L-fucose, $NAD^+$ regeneration is achieved via the reduction of lactaldehyde to 1,2-propanediol by lactaldehyde reductase (CRIB_1300), which was upregulated in the presence of L-fucose together with the L-fucose degradation gene cluster. In the spent medium of L-fucose grown cells, 1,2-propanediol was already seen at time point ∼8–10 h, whereas no lactate production was observed. Another way to regenerate $NAD^+$ is to reduce pyruvate to ethanol (Fig. 4). In the presence of both glucose and FOS, an upregulation was seen for the gene encoding the bifunctional aldehyde/alcohol dehydrogenase (CRIB_2231), which converts acetyl-CoA to ethanol. However, there were no samples in which ethanol was measured by HPLC analysis.

During growth on FOS, a small gene cluster (CRIB_601-CRIB_603) that includes a gene encoding an alternative sigma factor was significantly upregulated. This was also apparent in the control culture that was inoculated with FOS-preconditioned cells. This suggests that in the presence of FOS (or its derivatives sucrose or fructose) transcription is also regulated by RNA polymerase promoter recognition.

### Expression and regulation of other environmentally relevant functions in R. ilealis CRIB[T]

Noteworthy was the significant upregulation of a gene cluster related to iron transport (CRIB_892-CRIB-898) during growth on glucose and FOS compared to growth on L-fucose. The significance of this gene cluster for carbohydrate utilization is not known, however, several enzymes could be identified in the genome of *R. ilealis* CRIB[T] that use different forms of iron as cofactor, for example the hydrogenases involved in hydrogen metabolism (*Calusinska et al., 2010*), several ferredoxins, and the L-threonine dehydratase (CRIB-426) that was significantly upregulated during growth on L-fucose. As multiple transporters involved in the transport of iron compounds were predicted, it is possible that the uptake of iron provides a competitive advantage to other microbes that are dependent on iron for respiration and other metabolic processes (*Kortman et al., 2014*).

## Prevalence of *R. ilealis* in human datasets

*R. ilealis* was found to be a natural and abundant inhabitant of the rat small intestine, specifically of the ileum (*Gerritsen et al., 2011b*). To study its prevalence in humans, 16S rRNA amplicon sequencing datasets were investigated for the presence of *R. ilealis*-like 16S rRNA gene sequences. Unfortunately, with respect to composition analysis of human ileum samples, there is only a limited number of datasets available due to the sampling difficulties that are the result of the inaccessibility of this part of the intestinal tract. In the dataset published by *Alipour et al. (2016)*, a paediatric human dataset with samples from both healthy individuals and inflammatory bowel disease patients, we were not able to identify any *Romboutsia*-like 16S rRNA gene sequences. In the dataset published by *Gevers et al. (2014)*, one of the biggest 16S rRNA gene datasets published to date that includes samples obtained from multiple gastrointestinal locations (ileal and rectal biopsies and faecal samples) from both healthy individuals and inflammatory bowel disease patients, only a limited number of *R. ilealis*-like 16S rRNA gene sequences could be identified. In this dataset, the genus *Romboutsia* could be identified in two samples, with a relative abundance of 0.1% and 0.2%. In 173 cases the family *Peptostreptococcaceae* could be identified in these samples, but it was not possible to differentiate between the genera *Romboutsia* and *Intestinibacter*, due to 100% identity of their rRNA gene sequences in this region. The *Peptostreptococcaceae*-positive samples were obtained from both healthy and diseased individuals (including at least one with ileal overgrowth and a *Peptostreptococcaceae* abundance of 46%). It should be noted that both datasets contain only sequence data from paediatric ileal biopsy samples and therefore only mucosa-associated microbiota could be studied limited to a human population <17 years of age, which could explain the low prevalence of *R. ilealis*-like 16S rRNA gene sequences. Unfortunately, due to the limited number of available human datasets and the low prevalence of *R. ilealis*-like 16S rRNA gene sequences in ileal biopsy samples, it was not possible to find positive or negative correlations between prevalence and/or abundance of *R. ilealis* and specific human diseases.

## DISCUSSION

*Gerritsen et al. (2011b)* have shown by 16S rRNA gene sequence-based analysis that *R. ilealis* CRIB[T] is a dominant member of the small intestine microbiota in rats, especially in the ileum. The genomic and transcriptomic analysis of *R. ilealis* CRIB[T] reported here provides new insights into the genetic and functional potential of this inhabitant of the small intestine. Genomic analysis revealed the presence of metabolic pathways for the utilization of a wide array of simple carbohydrates in addition to a multitude of carbohydrate uptake systems that included a series of PTS systems, carbohydrate specific ABC transporters, permeases and symporters. This is in agreement with prior observations by *Zoetendal et al. (2012)*, who reported that the small intestinal microbiome is enriched for genes involved in the consumption of simple carbohydrates. However, small disagreements with prior observations were also observed. An enrichment for amino acid metabolism (*Zoetendal et al., 2012*) was not visible in *R. ilealis* CRIB[T], and considerable less COGs could be classified than for the average small intestinal bacterium (*Leimena et al., 2013*).

Since the small intestine is an environment in which environmental conditions change quickly due to the varying food intake of the host, microorganisms in this environment must be able to respond rapidly to such changes. As previously mentioned, the high number of rRNA operons found in the genome of *R. ilealis* CRIB[T] is an indication that this this strain is indeed able to adapt its metabolism quickly in response to changing conditions, as a high rRNA copy number has been associated with this trait (*Klappenbach, Dunbar & Schmidt, 2000*). Considering the small intestinal habitat, we chose to focus on key pathways involved in the utilization of specific diet- and host-derived carbon sources by whole-genome transcriptome analysis.

### Degradation of FOS and its possible role in cross-feeding

In the intestinal tract, the diet-derived carbohydrates that the host is unable to digest are important sources of energy for many microbes. In return, the host is dependent on the degradation of food-derived indigestible component by microbes for the release of certain essential metabolites (e.g., SCFA). Here we examined the growth of *R. ilealis* CRIB[T] on FOS, a relatively simple oligosaccharide that is indigestible by the host, and the metabolites that were released. The transcriptome of *R. ilealis* CRIB[T] grown on FOS was very similar to its transcriptome when grown on glucose, a monosaccharide used by the majority of microbes present in the intestinal tract. This is not surprising considering that glucose in addition to fructose is one of the two subunits present in FOS. Noteworthy was the accumulation of fructose in the culture supernatant during growth of *R. ilealis* CRIB[T] on FOS. Based on the genomic analysis there are no apparent reasons why fructose should not be metabolized as all the necessary metabolic enzymes are present. However, it has been previously observed that *R. ilealis* only grows weakly on D-fructose (*Gerritsen et al., 2014*). The absence of a fructose-specific transporter, which could be identified in close relatives that are able to grow on D-fructose, might explain the fructose accumulation during growth of *R. ilealis* CRIB[T] on FOS.

Differential gene expression analysis demonstrated the apparent FOS-induced upregulation of two separate gene clusters that were predicted to be involved in sucrose transport and degradation. However, based on the genomic analysis, no apparent pathways could be identified to be responsible for FOS degradation. A simple explanation for the observed growth on FOS could be extracellular degradation of FOS, followed by import of sucrose and/or glucose into the cell. Fructan degradation by extracellular enzymes is described for other (intestinal) microbes (*Van Hijum et al., 2006*). The observed accumulation of fructose during growth of *R. ilealis* CRIB[T] on FOS supports the hypothesis of extracellular degradation. However, no extracellular fructansucrase or glucansucrase could be predicted. Furthermore, no new candidates for this activity could be identified via the differential gene expression analysis described here. However, one possible candidate could be the predicted beta-fructofuranosidase present in the PTS system-containing sucrose degradation gene cluster. Next to the beta-fructofuranosidase-encoding gene, a gene was found to which no function could be assigned, but that was predicted to have a transmembrane region and a domain which could be involved in transport. Given that both loci overlap by a few nucleotides, and that the overlap is within a homopolymer

region, it is possible that both loci form one protein due to ribosomal slippage on the homopolymer (*Sharma et al., 2014*). This could possibly lead to an external membrane-bound enzymatically active protein, which would explain the accumulation of fructose. Future studies with mutant strains might shed more light on the specific contribution of the two predicted sucrose degradation gene clusters to the degradation of FOS, or even longer fructans (e.g., inulin), by *R. ilealis* CRIB[T]. Altogether, these results might indicate a possible role for *R. ilealis* CRIB[T] in intestinal cross-feeding networks by releasing D-fructose during growth on fructans like FOS, which can function as growth substrate for other microbes or be directly absorbed by the host.

### Fucose degradation and its advantages

Besides diet-derived carbohydrates, also host-derived carbohydrates are an important source of energy for some microbes. Unlike other members of the family *Peptostreptococcaceae*, *R. ilealis* CRIB[T] is able to grow on L-fucose, a predominantly host-derived carbon source (*Gerritsen et al., 2014*). The transcriptome analysis confirmed the presence of a functional L-fucose degradation pathway, similar to the pathways previously identified in other intestinal inhabitants such as *E. coli* (*Baldoma & Aguilar, 1988*), *Bacteroides thetaiotaomicron* (*Hooper et al., 1999*) and *Roseburia inulinivorans* (*Scott et al., 2006*). By gene sequence homology a similar pathway was found in *Clostridium perfringens* and the more closely related *C. sordellii* (Fig. S2). L-fucose is a common sugar present within the intestinal environment, since it is a monosaccharide that is an abundant component of many N- and O-linked glycans and glycolipids produced by mammalian cells, including the fucosylated glycans that are found at the terminal positions of mucin glycoproteins (*Becker & Lowe, 2003*). Fucosylated mucin glycoproteins are especially found in the (human) ileum (*Robbe et al., 2004*; *Robbe et al., 2003*). For both intestinal commensals and pathogens the ability to utilize L-fucose has been demonstrated to provide a competitive advantage in the intestinal environment (*Hooper et al., 1999*; *Stahl et al., 2011*). In *R. ilealis*, all enzymes for L-fucose degradation are present in one cluster, however, no fucosidase-encoding gene could be identified, which means that *R. ilealis* is not able to release L-fucose units from fucosylated glycans (e.g., mucin) by itself. Hence, in the intestinal environment *R. ilealis* is dependent on free L-fucose monosaccharides released by other microbes. Furthermore, a gene cluster involved in degradation of sialic acid (*Almagro-Moreno & Boyd, 2009*; *Vimr, 2013*; *Vimr et al., 2004*) was predicted from the genome, but no extracellular sialidase could be identified, which is similar to what has been found for *C. difficile* (*Ng et al., 2013*). This suggests that also for sialic acid, a common residue found in mucin glycoproteins, *R. ilealis* CRIB[T] seems to be dependent on the activity of other microbes. However, this also suggests that by its ability to use L-fucose and sialic acid monosaccharides, *R. ilealis* CRIB[T] is dependent for these host-derived sugars that are released by the action of extracellular enzymes of with mucus-degrading microbes like *B. thetaiotaomicron* or *Akkermansia muciniphila*. Besides niche competition with other commensals, fucose utilization may also be important in niche competition with pathogens. It was recently suggested that the host is able to regulate fucosylation of its intestinal epithelial cells in response to pathogen-induced stress and that microbes that are able to use fucose as an energy source

may contribute to the protection of the host against infections by endogenous pathogens (*Pickard et al., 2014*).

### Regulation of carbohydrate catabolism

In the intestinal environment *R. ilealis* CRIB$^T$ will encounter a wide array of carbohydrates that are either continually or transiently present. Prioritization of carbohydrate utilization is partly achieved at the transcriptional level by the selective expression of genes. The primary mechanism by which bacteria regulate the utilization of non-preferred carbohydrates in the presence of preferred carbon sources is known as carbon catabolite repression (CCR), a hierarchical system for coordinating sugar metabolism (*Deutscher, 2008*). The fact that, compared to glucose and FOS, L-fucose is utilized by a pathway that does not directly involve fructose-1,6-bisphosphate, a key metabolite in the regulation of CCR of Gram-positive bacteria, made it possible to study CCR by either glucose or FOS. The transcriptome analysis suggests that some genes and operons in *R. ilealis* CRIB$^T$ were indeed subject to CCR in response to the presence of glucose. For example, two gene clusters predicted to be involved in hexuronate metabolism (CRIB_649-CRIB_652 and CRIB_2244-CRIB_2249), pathways that make the use of D-glucuronate and D-galacturonates as sole carbon source possible, were significantly upregulated during growth in the presence of L-fucose compared to growth on glucose (Table S5). In addition, the gene cluster predicted to be involved in sialic acid utilization (CRIB_613-CRIB_616) was downregulated in the presence of glucose as well. Furthermore, when comparing the expression of the gene cluster involved in L-fucose degradation during growth on glucose relative to the growth in the absence of a carbon source (control condition), this gene cluster appeared to be under CCR as well, in the presence of glucose (Table S5). These results suggest that in *R. ilealis* CRIB$^T$, multiple gene clusters that are involved in the use of alternative carbon sources are subject to CCR.

### Expression and regulation of niche-specific functionalities in *R. ilealis* CRIB$^T$

Microbes residing in the intestinal tract have to withstand the harsh environmental conditions specific for the intestine. In this context, it was interesting that we identified a urease gene cluster in *R. ilealis* CRIB$^T$ (CRIB_1381-CRIB_1388), expression of which appeared to be induced in carbon source limiting circumstances. The fact that this gene cluster was significantly upregulated when grown in the absence of an additional carbon source compared to growth on glucose, possibly suggests CCR of the urease gene cluster. However, upregulation of this gene cluster in the absence of an exogenous carbon source might also be a possible mechanism. Urea in the intestinal tract is derived from the breakdown of amino acids. *Helicobacter pylori* is a well-known example where urease activity contributes to the survival of the bacterium in the acidic environment of the stomach (*Marshall et al., 1990*). For some of the urease-positive bacteria, this enzyme has been shown to act as a virulence factor as it is responsible for urea hydrolysis that leads to increased pH and ammonia toxicity (*Rutherford, 2014*). However, for commensal intestinal bacteria ureases can probably function as colonization factors as well, as they contribute in general to acid resistance and thereby play a role in gastrointestinal survival (*Marshall et al., 1990*). Urea is released into all parts of the intestinal tract via diffusion from the

blood, but it has been reported that pancreatic excretions and bile are a main route of entry (*Bergner et al., 1986*). So far, we have not been able to demonstrate urease activity in *R. ilealis* CRIB[T] (*Gerritsen et al., 2014*). However, different mechanisms for the expression of urease have been identified in other microbes: constitutive, inducible by urea, or controlled by nitrogen source availability (*Mobley, Island & Hausinger, 1995*). For *C. perfringens* for example, the urease activity, which is plasmid borne, was shown to be only expressed in nitrogen-limiting conditions (*Dupuy et al., 1997*). The increased urease gene expression by *R. ilealis* CRIB[T] observed in the control condition, in the absence of an additional carbohydrate, suggests an alternative mechanism for regulation of urease gene expression.

## CONCLUSIONS

We are just starting to elucidate the composition and function of the microbial communities in the mammalian small intestine. Recently we have reported the isolation and characterization of *R. ilealis* CRIB[T] from the small intestine of a rat (*Gerritsen et al., 2014*). In rats, this species was identified to be a dominant member of the ileal microbiota (*Gerritsen et al., 2011b*). Here we applied a holistic systems biology approach, involving several fields of experimental and theoretical biology, to study *R. ilealis* CRIB[T]. In conclusion, *R. ilealis* CRIB[T] is a strain that is able to utilize an array of carbohydrates using different and partially redundant pathways. Its ability to use host-derived sugars that are liberated by other microbes suggests that *R. ilealis* CRIB[T] is dependent on mucus-degrading microbes, like *B. thetaiotaomicron* or *A. muciniphila*. In contrast, it has only limited ability to *de novo* synthesize amino acids and vitamins, and hence the organism shows an adaption to a nutrient-rich environment in which carbohydrates and exogenous sources of amino acids and vitamins are abundantly available. In addition, we were able to pinpoint potential mechanisms that might enable this organism to survive in the competitive small intestinal environment. These mechanisms include bile salt hydrolase and urease enzymes, which enhance the organism's ability to handle in particular small-intestinal conditions.

It has to be emphasized that the results presented in this study correspond to one specific strain and that different strains belonging to the same species could possibly encode for different functions, including utilisation of specific glycans as previously described by *Crost et al. (2013)*. However, a deeper investigation of key players in the intestinal tract like *R. ilealis* CRIB[T] and others will lead to a better understanding of how the microbial communities in us function as a whole. The more we understand how each organism works, and how they interact, the better we get an insight into these environments and can predict how nutrition will influence our health and well-being.

## ACKNOWLEDGEMENTS

The authors thank Hans Heilig (Wageningen University, Laboratory of Microbiology) for his help with genomic DNA isolations and Jasper Koehorst (Wageningen University, Laboratory of Systems and Synthetic Biology) for his help with the genome annotation.

### Funding

This work was supported by a grant of SenterNovum (Agentschap NL), an agency of the Dutch Ministry of Economic Affairs (FND-07013). B Hornung is supported by Wageningen University and the Wageningen Institute for Environment and Climate Research (WIMEK) through the IP/OP program Systems Biology (project KB-17-003.02-023). The funders had no role in study design, data collection and analysis, decision to publish, or preparation of the manuscript.

### Grant Disclosures

The following grant information was disclosed by the authors:
SenterNovum (Agentschap NL): FND-07013.
Wageningen University.
Wageningen Institute for Environment and Climate Research (WIMEK).

### Competing Interests

J Gerritsen is currently an employee of Winclove Probiotics. However, these associations do not influence the objectivity, integrity and interpretation of the results that presented in this manuscript. Sacha A.F.T. van Hijum is an employee of NIZO, Kernhemseweg 2, 6718 ZB, Ede, the Netherlands and Vitor A.P. Martins dos Santos is an employee of LifeGlimmer GmbH, Markelstrasse 38, Berlin, Germany. Willem M. de Vos and Hauke Smidt are Academic Editors for PeerJ.

### Author Contributions

- Jacoline Gerritsen conceived and designed the experiments, performed the experiments, analyzed the data, wrote the paper, prepared figures and/or tables, reviewed drafts of the paper.
- Bastian Hornung performed the experiments, analyzed the data, wrote the paper, prepared figures and/or tables, reviewed drafts of the paper.
- Bernadette Renckens performed the experiments, analyzed the data, reviewed drafts of the paper.
- Sacha A.F.T. van Hijum, Vitor A.P. Martins dos Santos and Peter J. Schaap analyzed the data, reviewed drafts of the paper.
- Ger T. Rijkers, Willem M. de Vos and Hauke Smidt conceived and designed the experiments, analyzed the data, reviewed drafts of the paper.

### DNA Deposition

The following information was supplied regarding the deposition of DNA sequences:
All related data has been deposited at the European Nucleotide Archive. The raw reads for the genome can be accessed via the accession numbers ERR366773, ERX397233, ERX397242 and ERX339449. The assembly can be accessed under LN555523–LN555524. The RNAseq data has been deposited under the numbers ERS533849–ERS533860.

## Data Availability

All data has been submitted to the EBI.

## Supplemental Information

Supplemental information for this article can be found online at http://dx.doi.org/10.7717/peerj.3698#supplemental-information.

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
