# Peer review of "Genomic and functional analysis of Romboutsia ilealis CRIBT reveals adaptation to the small intestine"

_PeerJ, doi:10.7717/peerj.3698_

## Round 0.1 · original submission · Minor Revisions

· Academic Editor

Minor Revisions

The paper by Gerritsen et al is clear and well written. It describes genome and functions of a type strain that corresponds to the new species Romboutsia ilealis that is a dominant member of the microbiota in rat ileum. Findings and explanations are robust and the study constitutes a valuable contribution to the field.

I only have a few, minor comments:

- Authors might consider slight revisions in their manuscript to improve the understanding by the reader (see Reviewers comments).
- Since the CRIB isolate is the only available isolate to date I wonder if authors could mine for closely related sequences in (potentially) available 16S repertoire or shotgun metagenomics datasets? This could maybe bring additional genomic information (shotgun datasets), and/or highlight positive or negative association with specific health conditions (16S & shotgun datasets)?
- In addition, since this species was associated with improved health status in acute pancreatitis in their previous papers, one could expect this point to be discussed. Could this association be explained by specific functions encoded by the genome? And/or by cross-feeding with other species that were positively associated with improved health condition?
- As suggested by Reviewer 3, I would also include in the discussion the fact that these results correspond to one specific strain and that different strains could possibly encode for different functions, including utilisation of specific glycans as previously described by Crost et al (Utilisation of mucin glycans by the human gut symbiont Ruminococcus gnavus is strain-dependent. 2013. PLoS ONE 8:e76341)

Overall a very interesting study that deserves publication in PeerJ.

·

Basic reporting

The findings have been reported in a clear and rationale manner with well-detailed references to the published literature, but I found a few grammatical errors in English (See attached PDF with annotations), which should be corrected.
At the following places, the manuscript can be improved by providing more clarity:
Line 49-50- “Uncovered potential mechanisms for competition with mucus-degrading microorganisms”- Since the authors write in Line 492-493 that no fucosidase gene was identified in R.ileas, it is unclear whether it relies on the mucus-degrading microbes to obtain fucose as its energy source or competes with them. Therefore, in the abstract it is not right to make the above statement. A fusose degradation pathway was identified which may provide this microrganism a competitive edge over other microbes in the intestine is a more accurate description of the finding. It is also unclear whether the author want to suggest a symbiotic relationship with the mucus degrading microbes.
Line 103- Same critique as above (for Line 49-50)
Line 185-190- Authors describe that R.ileas has the highest number of rRNA copies reported upto date but do not describe how many of these are present in the original gene cluster and how many are present outside it and be clear on whether they think that the gene duplication is relevant to any physiological function in the intestine.
Line 191- Please give a full form for ‘COG’ or mention that the genome was searched against COG database of NCBI.
Line 256-275- It is unclear where do the authors first describe that they identified the gene cluster in involved in fucose degradation?
Table 1- Please describe in the legend about what do question marks in the figure indicate- genes not identified in R.ileas or not known for this pathway?
Table 2- Please describe the units in which fermentation products are reported either in the table or the legend.

Experimental design

No Comments

Validity of the findings

In discussion, can you compare your findings to some extent with what has been shown with small intestine of humans (as shown by Zoetendal, Leimena et al)?

Additional comments

The authors present a detailed study of an important player in rat’s small intestine microbiota. They sequenced, analyzed, annotated the genetic elements of this microorganism and further validated the results through transcriptome analysis after growth on various carbohydrate containing media. The findings are important step to understanding the gut microbiome and it's effect on host health.

Reviewer 2 ·

Basic reporting

No comments

Experimental design

No comments

Validity of the findings

No comments

Additional comments

The study of “Genomic and functional analysis Romboutsia ilealis CRIB reveals adaptation to the small intestine” is very interesting as well as important to understand the complex microbiota in the small intestine. It seems the authors are the first one to report the detailed genomic analysis of this organism. The study is well designed and executed scientifically and presented nicely. The following points should be considered to improve the manuscript,

1. The authors need to give more insight about the IBD and its relationship with this bacteria
2. Although this bacteria is derived from rat, the authors need to explain the implication of the study to human small intestine?
3. They can include close homologs of this bacteria based on the genome comparison.

·

Basic reporting

Gerritsen et al present a well written and aesthetic manuscript characterising the little studied gastrointestinal dwelling Romboutsia ilealis. It is an interesting article detailing the metabolic capacity of the organism as studied in the lab and further putative metabolic processes as predicted from sequence data.

All raw data is shared and available online.

Experimental design

The aims are clearly stated in the introduction and research question well articulated. Methods are well described and investigations are conducted to a high standard.

Validity of the findings

The data is robust and conclusions well stated. I would draw back from using the term adaptation as frequently as the authors do - as only one isolate has been studied.

Additional comments

Major concerns
• I am not comfortable with the use of the term ’adapted’ throughout the manuscript. I don’t think you can draw species-level conclusions from a single sequenced isolate.
• I would have liked to have seen a phylogenetic tree of the isolate alongside some other (closely related) gastrointestinal dwelling bacteria.
• You mention that there are a lot of unidentified genes – does this explain some of the ‘missing’ genes some of the amino acid/metabolic pathways you mention?

Minor concerns
• Line 38: change to ‘growth’
• Line 40: remove ‘hence’;
• Line 46: …pinpointing *of* the key components…
• Line 92: remove ‘completely’
• Lines 99-104: move to discussion
• Lines 212-220: How does this compare with the growth you observed in the lab?
• Lines 230, 274, 515 and z: ‘data not shown’ – just include as supplementary
• Line 329: re-phrase ‘growth was slightly less..’
• Line 397: remove ‘remarkable’.
• Line 408: remove ‘interestingly’
• Lines 412-435: discussion?
• Line 456: remove ‘remarkable’.

·

Basic reporting

This manuscript has many strong points to suggest it. The work is clearly professional and has been carried out thoroughly and with a combination of analyses and techniques that speaks to completeness and experience.

That being said, in the writing there are several structural weaknesses as well as small errata that undermine the many strengths of this work. All of these are addressable with some heavy editing, but on the flip side, there is opportunity to improve the impact and readability of this article. Personally, I found this article very exciting, but oppositely frustrating because I found some of the most interesting portions to read somewhat confusingly. Below, I highlight some of the areas where improvements might be made. These represent general themes that can be applied throughout the manuscript to more than the examples I list:

1) Repetitious statements

There are places where the writing mimics itself in close proximity which makes it feel repetitious. For example Lines 55-57 and Lines 69-71:

"This has lead to a complex network of host-microbe and microbe-microbe interactions in which the intestinal microbes and the host co-metabolise many substrates."

"The wide array of microbial genes present in the intestinal tract in addition the host's own genome provides insight into the complex network of possible host-microbe and microbe-microbe interactions."

and Lines 91-92 and Line 97:

"Here we describe..."

"Here we describe..."

takes me to the concluding summary of the introduction twice

2) Paragraph subjects

Lines 91-92 and Line 97 are also indicative of another feature that occurs in the manuscript which is failure to identify the subject of a paragraph. I would argue that the "Here we describe ..." wording is, in fact, deserving of it's own paragraph at the end of the introduction, but the two examples mentioned here are in different paragraphs.

Another example is in the Discussion section starting with Line 448. This paragraph begins with "carbohydrates ... are important sources of energy for many microbes" goes on to discuss FOS and then goes into details about the pathway. This part is fine. The next portion which then goes into possible mechanisms of the fructose accumulation and the different scenarios and evidence should begin with a clear statement of question, then presentation of the evidence (perhaps in a separate paragraph). Currently it reads as a list of "maybe it's this" or "maybe it's that" without focus or conclusion. It would orient the reader to zoom out a little before zooming into the details with things like, why is this important?

3) Random facts

There are, on occasion, some random facts scattered throughout the manuscript that I cannot place in the larger picture. For example, in Lines 90-91 "analysis on Streptococcus isolates of small intestinal origin has shown that these microbes are highly adapted to a highly dynamic environment" does not appear to have a purpose. As far as I can tell, the purpose of citing this work might be better illustrated by a sentence more along the lines of "Other genomic studies have indicated environment-specific adaptations to the small intestine in the form of ... which was evidenced by ..."

As another, more subtle example, Line 213 begins a detailed argument on the potential source of propionate that is not given at the same level as the rest of the paragraph which discusses the overall metabolic characteristics identified on a very course-grained level. This jumping from broad picture to minute detail without a segue ends up having the effect of inserting a random detail where it does not belong. The segue could have been something along the lines of, "The only unaccounted metabolite identified was ... which was noteworthy for the following reasons ... However, there are some possible explanations we can offer ..." Which also suggest that this might be better off in the discussion in some sense.

4) Start with the question/subject of discussion. I.e., why is the next fact you are going to give me interesting?

Discussion section in general should start each paragraph with what the question is that is being addressed. Perhaps a hint at where the paragraph is leading to would be appropriate to put nearer to top of each paragraph.

Similarly in the "Additional genes of ecological interest" section, these do not appear to have "ecological interest" as stated. It would be important to identify why urea and bile acid metabolism matter. For bile acid metabolism, there are any number of reviews linking secondary bile acid metabolism to colon cancer and GI other diseases, though I am (of course) most familiar with colon cancer given my specialty. Some reviews can be taken from our work in colon cancer:

Hale, Vanessa L., et al. "Shifts in the fecal microbiota associated with adenomatous polyps." Cancer Epidemiology and Prevention Biomarkers 26.1 (2017): 85-94.

5) Other mentions:

a) Continuity: For example, Line 74 has "However, we have only limited understanding of the spatial and temporal..." for which one would think there was a line beforehand stating something like "The spatial and temporal ... are important to ...". Instead, the line before the "However" reads, "To be able to maintain themselves in an ecosystem such as the intestinal tract, microbes have adapted or even specialized in foraging certain niche-specific substrates" which seems to have very little to do with spatial and temporal, at least explicitly.

b) Lines 126-128: This seems like a lot of methods that are very central to the subject of the manuscript and seems a bit odd to shove them all off to supplemental without at least some information in the main text. Similarly, the transcriptional analysis is very detailed in the main text. It would be better to have a high-level summary of all methods in main text and then a detailed summary of all methods in supplemental.

c) "Multiple amino acid transporters" is an interesting result. We had also identified a gut microbe that was heavy in transporter dependencies in our own work and it might be worth a comparison to see if there are similar trends.

Jeraldo, Patricio, et al. "Capturing One of the Human Gut Microbiome’s Most Wanted: Reconstructing the Genome of a Novel Butyrate-Producing, Clostridial Scavenger from Metagenomic Sequence Data." Frontiers in Microbiology 7 (2016).

d) Order: Lines 265-270. Flipping the order of these two sentences would give better reasoning to the lines. "No separate pathway for D-arabinose could be predicted" would matter more if the reader knew why it mattered at all. The point is that there was a transporter, but no pathway, so I would start with we identified a transporter, then however, no pathway was detected. This would make more sense as to why you were even mentioning a D-arabinose pathway that you could not fine. There are otherwise all kind of things you cannot find, but they do not all matter.

Experimental design

This article is relevant and appropriate for publication in PEERJ. The research question is relevant and meaningful, though in places not well-defined. However, as this is mostly due to the structure of the writing, I would clarify that I find the experimental design strong with only a few minor comments. The methods are fairly rigorous and the analysis is fairly extensive.

The two minor comments I have are:

1) In describing the metabolic modeling, there is no mention of media conditions in the main text. In order for the paper to be self-contained, it would be important to understand if the pathway analysis was performed on complete media, with or without oxygen, or else how the conditions of the small bowel were mimicked through use of the media conditions.

2) With regard to the use of COG (which btw, requires a reference), there are many other alternatives such as PATRIC, RAST, etc.., Why was one of the alternatives not used?

Small note: It should be noted that 70% identification of gene functions is actually fairly usual. This is more or less in line with expectations from a complete genome.

Validity of the findings

Data is robust, statistically sound, and controlled. Conclusions are important and linked to original research question with support. Speculations are well developed and well identified. The only weaknesses here are identified from the structure of the writing and overall the work in this manuscript is very strong and would be appropriate for PEERJ provided some edits to the writing.

Additional comments

Despite the rather lengthy writing critique, I would like to re-emphasize the strength and impact of this work. This manuscript contains some very interesting scientific findings that should be published. I hope some of the remarks may be used to allow the strengths of the work to be appropriately highlighted in by the final article.

---

## Round 0.2 · accepted · Accept

· Academic Editor

Accept

The authors answered all minor comments made by reviewers, thus bringing additional information and improving the readability of the manuscript. The manuscript can consequently be published in its current form.